# Ready, Set, Plan! Planning to Goal Sets Using Generalized Bayesian Inference

**Jana Pavlasek**
University of Michigan
pavlasek@umich.edu

**Stanley Robert Lewis**
University of Michigan
stanlew@umich.edu

**Balakumar Sundaralingam**
NVIDIA
balakumars@nvidia.com

**Fabio Ramos**
NVIDIA
University of Sydney
ftozetoramos@nvidia.com

**Tucker Hermans**
NVIDIA
University of Utah
tucker.hermans@utah.edu

**Abstract:** Many robotic tasks can have multiple and diverse solutions and, as such, are naturally expressed as goal sets. Examples include navigating to a room, finding a feasible placement location for an object, or opening a drawer enough to reach inside. Using a goal set as a planning objective requires that a model for the objective be explicitly given by the user. However, some goals are intractable to model, leading to uncertainty over the goal (e.g. stable grasping of an object). In this work, we propose a technique for planning directly to a set of sampled goal configurations. We formulate a planning as inference problem with a novel goal likelihood evaluated against the goal samples. To handle the intractable goal likelihood, we employ generalized Bayesian inference to approximate the trajectory distribution. The result is a fully differentiable cost which generalizes across a diverse range of goal set objectives for which samples can be obtained. We show that by considering all goal samples throughout the planning process, our method reliably finds plans for manipulation and navigation problems where heuristic approaches fail.

**Keywords:** Planning as inference, variational inference, nonparametric learning.

## 1 Introduction

In order to accomplish diverse tasks in unstructured environments, robots need the ability to generate motion plans to different user-specified goals. Such goals are often naturally expressed as *goal regions* in the workspace, rather than single point goals (e.g. navigate to a certain room, place an object on a shelf). Traditional trajectory planning techniques require the goal regions to be explicitly specified as planning objectives. This specification is challenging in cases where the goal region is *uncertain*. Examples of uncertain goal regions include those which are intractable to model explicitly, such as stable grasp poses across diverse objects [1, 2] or user preferences [3, 4].

We seek to solve planning problems under goal region uncertainty given only a fixed set of goal demonstrations. The key insight of our approach is to model these demonstrations as samples from an underlying goal distribution. Our *goal set planner* plans directly to these goal samples, resulting in a generalizable, data-driven planning objective which can be employed across goal regions, eliminating the need to explicitly model the goal. Planning to goal distributions has been formulated as a flexible representation for a number of objectives [5], but requires a fully specified parametric goal distribution. Our approach instead treats the goal as an *implicit distribution*, from which we can obtain samples (e.g. successful grasp poses [6]) but which cannot be evaluated explicitly. We formulate this problem as inference-based planning [7, 8, 9], and seek a distribution of trajectories which terminate in high-density regions of the goal distribution (see Figure 1). Due to the complex,

7th Conference on Robot Learning (CoRL 2023), Atlanta, USA.

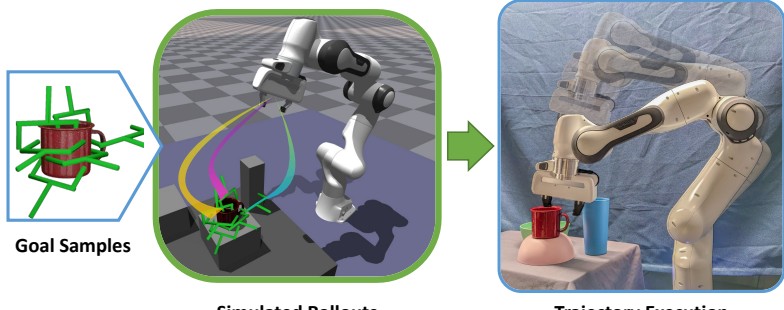

**Goal Samples**  **Simulated Rollouts**  **Trajectory Execution**

**Figure 1:** Our *goal set planner* considers a discrete set of goal samples as the planning objective. By considering multiple possible goals within the trajectory optimization process, we can handle challenging environments with clutter and dynamic obstacles.

multi-modal nature of the planning problem, we consider particle-based trajectory distributions and perform inference using Stein Variational Gradient Descent [10, 11]. In contrast to previous data-driven approaches to intractable planning objectives [1, 2, 3, 4, 6, 12, 13], we require no upfront learning before planning begins.

**Primary contributions.** We formulate the problem of goal set planning as *generalized Bayesian inference* (GBI), an emerging statistical technique which enables inference for intractable likelihoods. GBI refers to a class of problems in which the likelihood function is specified through a generic loss, typically a divergence between two distributions evaluated against observations [14, 15, 16]. We demonstrate that divergences can be computed over implicit goal distributions, which allows us to leverage a loss function that quantifies the divergence between the terminal state samples and the goal samples to infer the posterior trajectory distribution [17]. We then propose a Stein variational inference methodology to solve the problem as a sequence of gradient updates of a set of trajectories.

**Secondary contributions.** We present a number of potential loss functions which can be applied within this framework. We additionally compare to several previously proposed heuristic approaches [5, 18]. We demonstrate the applicability of our approach in planar navigation and in a high-dimensional grasping task. By considering the full set of goal samples throughout the planning process, our planner reliably finds reachable grasps in scenarios where heuristic approaches fail to reach their selected pose from the goal set. Additional demonstrations and data are available at https://janapavlasek.com/projects/goal-sets.

## 2 Related Work

Trajectory planning to a goal can be stated as an optimization problem with the objective of finding a minimum cost path to a point in the robot workspace and can be efficiently solved using differentiable costs and dynamics [19, 20, 21]. Casting planning as probabilistic inference has become a popular technique for achieving robust control under uncertainty [22, 11, 23, 9, 24]. One popular approach is to employ a nonparametric, particle-based representation of the trajectory distribution, iteratively updating the posterior through importance sampling [23, 9]. Other methods use variational inference to efficiently infer the trajectory distribution [22, 11, 25].

Most related to our work is that of Conkey and Hermans, who formulate trajectory planning to goals defined as a probability distribution [5]. Our proposed approach also considers the goal to be a distribution, but we represent the goal as an implicit (sampled) distribution [17] eliminating the need for the distribution to be explicitly specified by the user. The problem of planning to a set of goals has been considered by the manipulation community for robotic grasping. A popular approach for grasp planning to a goal set has been to formulate trajectory optimization and grasp selection as independent steps [26, 27, 18]. These methods first select a single goal from the set prior to planning based on a scoring metric, then treat the selected sample as a point goal. Similar approaches have been used to heuristically guide sampling-[28] and graph-based [29] motion planners to goal samples from sets in other contexts. Another approach is to jointly perform trajectory optimization and grasp

selection. Dragan et al. [30] add a manually defined goal set constraint to the problem of trajectory optimization. Wang et al. [31] perform grasp selection and refinement online by estimating the grasp distribution. These methods require domain-specific information about the grasping problem and do not generalize across planning problems. Data-driven approaches to represent intractable planning objectives either employ high-fidelity simulation to measure success (e.g of grasps) [1, 2, 6, 13], or demonstrations to represent user preferences [3, 4, 12]. In these techniques, an upfront computational cost is required before planning begins, and must be performed again whenever environmental conditions, such as observability or dynamic environments, necessitate replanning.

## 3  Goal Sample Set Planning as Inference

Given a robot with state $x_t \in \mathcal{X}$ at time $t$, we consider the problem of planning to a *goal set* within the state space, $\mathcal{G} \subset \mathcal{X}$. We consider the case of arbitrarily complex goal regions $\mathcal{G}$ that do not admit explicit analytic specification. Instead we can obtain a set of valid goal samples, $\tilde{G} = \{x_g^{(i)}\}_{i=1:N}$, $x_g^{(i)} \in \mathcal{G}$. Examples of this type of goal include data labelled by simulation or human demonstration.

We seek a policy $u_t = \pi(x_t)$ such that the terminal state after a time horizon $T$ lies within the goal region, $x_T \in \mathcal{G}$, given only goal samples $\tilde{G}$. The robot trajectory is defined as a sequence of states and actions, $\tau = \{(x_t, u_t)\}_{t=0:T}$. We seek the minimum cost trajectory, $\tau^*$, which solves:

$$\tau^* = \arg\min_{\tau}  C\left(x_T; \tilde{G}\right) + \sum_{t=0}^{T-1} c_t(x_t, u_t; z), \tag{1}$$

where $z$ denotes the environmental observation (e.g. the map of the environment). The running cost, $c_t(x_t, u_t; z)$, is chosen based on the domain (e.g. quadratic cost, obstacle cost, etc.). Our primary contribution is the formulation of the terminal cost, $C(x_T; \tilde{G})$, when the goal region is represented only by a set of samples, $\tilde{G}$.

The planning objective in Eq. 1 can be restated in terms of probabilistic inference using the formalism of planning as inference [7, 8, 22, 9]. In planning as variational inference, we seek to find a distribution over trajectories $q(\tau) \in \mathcal{Q}$ that minimizes the divergence with the posterior distribution of trajectories, $p_O(\tau \mid z)$, defined from the cost in Eq. (1):

$$q^*(\tau) = \arg\min_{q \in \mathcal{Q}} D_{KL}\left(q(\tau) \,\|\, p_O(\tau \mid z)\right). \tag{2}$$

We can factorize the posterior over trajectories, $\tau$, given an environment observation, $z$, and a set of goal samples, $\tilde{G} = \{x_g^{(i)}\}_{i=1:N}$, as:

$$p_O(\tau \mid z, \tilde{G}) \propto p_{\text{goal}}(\tilde{G} \mid x_T) \prod_{t=1}^{T-1} p(z \mid x_t) \prod_{t=1}^{T-1} p(x_{t+1} \mid x_t, u_t) \tag{3}$$

$$= p_{\text{goal}}(\tilde{G} \mid x_T) p_O(z \mid \tau') p(\tau'), \tag{4}$$

where $\tau' = \{(x_t, u_t)\}_{t=0:T-1}$ denotes the portion of the trajectory up to time $T - 1$, $x_T$ denotes the terminal state of the trajectory, and $p_{\text{goal}}(\tilde{G} \mid x_T)$ is the goal likelihood given the implicit goal distribution $\tilde{G}$. We model the likelihood $p_O(z \mid \tau')$ in terms of the trajectory cost yielding:

$$p_O(z \mid \tau') \approx \frac{1}{Z} \exp\left(-\alpha \sum_{t=0}^{T-1} c_t(x_t, u_t; z)\right), \tag{5}$$

where $Z$ is a normalization term. The likelihood $p_{\text{goal}}(\tilde{G} \mid x_T)$ in Eq. (4) is intractable due to the implicit goal distribution. The following section addresses the key technical question of this work: how can we perform inference on the intractable posterior arising from an implicit goal distribution?

## 4  Planning to Implicit Goal Distributions as Generalized Bayesian Inference

We propose applying generalized Bayesian inference to directly approximate the solution of Eq. (4). While one could fit a parametric model of $\hat{p}_{\text{goal}}(\mathcal{G} \mid x_T)$ to the sampled data $\tilde{G}$, we seek a generic

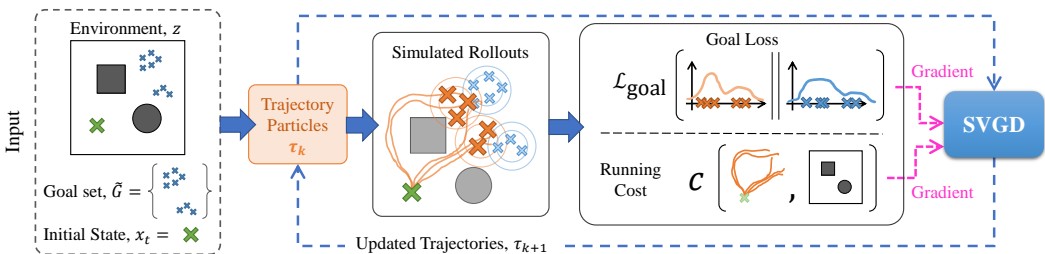

**Figure 2:** Our SVGD-based planner takes as input a set of goal samples, $\tilde{G}$, an environmental observation, $z$, and an initial state, $x_0$ and iteratively updates a set of trajectory particles, $\tau$. At each iteration, the particles are rolled out using a simulator. Each rollout is then evaluated in terms of its running cost, $c$. A goal loss , $\mathcal{L}_{\text{goal}}$, computes the discrepancy between the terminal points of the trajectory distribution and the goal set. SVGD updates the trajectory particles using the gradients of the combined running and goal losses.

objective for planning directly to the implicit distribution that avoids task-specific pre-processing. We now give a brief overview of generalized Bayesian inference and Stein variational inference before explaining how we combine them to solve the goal set planning problem. Figure 2 visualizes our approach to planning as generalized Bayesian inference using SVGD.

**Generalized Bayesian Inference.** Generalized Bayesian inference (GBI) accounts for intractable inference problems arising from evaluating a loss function instead of a likelihood that factorizes over each observation [16]. Given a posterior $p(x \mid \mathbb{Y})$, where $x$ are arbitrary random variables and $\mathbb{Y} = \{y^{(i)}\}_{i=1:N}$ is a set of $N$ observed data points, GBI approximates the posterior as follows:

$$p_{\mathcal{L}}(x \mid \mathbb{Y}) \propto p(x) \exp\left(-\beta \mathcal{L}(x, \mathbb{Y})\right) \qquad (6)$$

where $\mathcal{L}(x, \mathbb{Y})$ defines a loss between the random variables and observed data. Typically, the loss measures divergence (e.g. KL) between the data and query variables. The posterior $p_{\mathcal{L}}(x \mid \mathbb{Y})$ recovers the conventional Bayesian posterior when $\beta = 1$ and $\mathcal{L}(x, \mathbb{Y}) = -\sum_{i=1}^{N} \log p(y^{(i)}|x)$. We can compute the approximate posterior by solving the following variational optimization problem:

$$q_{\mathcal{L}}(x \mid \mathbb{Y}) = \operatorname*{arg\,min}_{q \in \mathcal{Q}} \beta \, \mathbb{E}_{x \sim q}\left[\mathcal{L}(x, \mathbb{Y})\right] + D_{KL}\left(q(x) \,\|\, p(x)\right) \qquad (7)$$

**Stein Variational Inference.** We build on work that approximately solves Eq. (2) using Stein variational inference [11, 32]. Stein variational inference represents the candidate distribution $q(\tau)$ nonparametrically by a set of particles, each representing a trajectory, $\{\tau^{(i)}\}_{i=1:M}$. Stein variational gradient descent (SVGD) [10] employs gradient-based optimization over the particle set to minimize the kernelized Stein discrepancy [33] between the true and the approximate posterior. SVGD can solve high-dimensional planning problems involving complex, multi-modal posteriors [11, 32]. SVGD is an iterative algorithm which applies the following update to each particle $i$ at iteration $k$:

$$\tau_k^{(i)} \leftarrow \tau_{k-1}^{(i)} + \gamma \phi(\tau_{k-1}^{(i)}) \qquad (8)$$

$$\phi(\tau) = \frac{1}{M} \sum_{j=1}^{M} k(\tau^{(j)}, \tau) \nabla_{\tau^{(j)}} \log p_O(\tau^{(j)} \mid z) + \nabla_{\tau^{(j)}} k(\tau^{(j)}, \tau) \qquad (9)$$

where $k(\tau^{(j)}, \tau)$ is a kernel function between trajectories. We can interpret the first term inside the summation of Eq. (9) as an attractive force that moves particles according to the gradient of the log-posterior, while the second term acts as a repulsive term keeping particles from collapsing to a single point estimate. Thus SVGD leverages parallel gradient-based optimization to generate a diverse set of samples more efficiently than Markov chain Monte Carlo (MCMC) samplers [9].

**Terminal Losses for Goal Sets.** We leverage results from the GBI literature which approximate the optimization problem in Eq. (7) using a divergence metric between two implicit distributions [17]. We define the implicit terminal distribution as $\tilde{X}_T = \{x_T^{(i)}\}_{i=1:M}$ and solve:

$$q_{\mathcal{L}}(x_T \mid \tilde{G}) := \operatorname*{arg\,min}_{q \in \mathcal{Q}} \mathcal{L}_{\text{goal}}(\tilde{X}_T, \tilde{G}) \qquad (10)$$

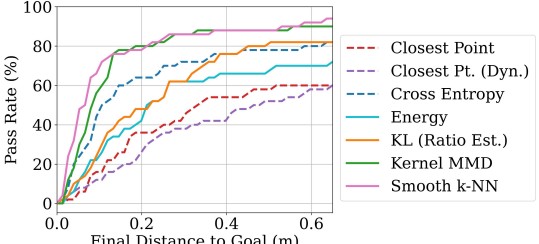

**Figure 3:** Pass rate for different distance thresholds between the final state and the nearest goal sample. Dashed lines represent baseline methods.

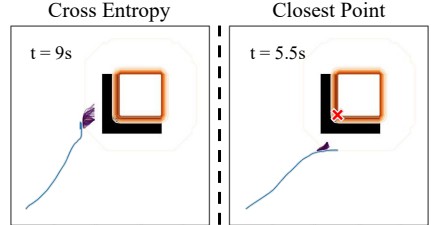

**Figure 4:** Cross Entropy (left) and Closest Point (right) baselines fall into local minima failing to reach the goal. The trajectory is shown in blue, the trajectory distribution in purple, and the goal distribution in orange contours. The red 'x' shows the closest point.

The loss function $\mathcal{L}_{\text{goal}}(\tilde{X}_T, \tilde{G})$ measures the discrepancy between the terminal state distribution and the distribution of goal states, represented implicitly by samples $\tilde{X}_T$ and $\tilde{G}$ respectively.

We note that the loss defined in Eq. (10) provides a measure over the set of terminal states, rather than an individual likelihood per trajectory particle required by Eq. (4). A key insight of our approach is to solve the resulting planning as inference task with SVGD, which employs the gradient of the particle likelihood, without the need to explicitly evaluate the goal loss per particle. We define the gradient in Eq. (9) as follows, combining the results from Eq. (4) and Eq. (10):

$$\nabla_{\tau^{(j)}} \log p_O(\tau^{(j)} \mid z) = \nabla_{\tau^{(j)}} \log p_O(z \mid \tau^{(j)}) + \nabla_{\tau^{(j)}} \log p(\tau^{(j)}) - \beta \, \nabla_{\tau^{(j)}} \mathcal{L}_{\text{goal}}(\tilde{X}_T, \tilde{G}) \quad (11)$$

The terminal loss must be differentiable with respect to the trajectory samples and efficient to compute so it can be used in a control loop. We consider the following *two sample* tests which meet this criteria: The kernel Maximum Mean Discrepancy (MMD) [34], a KL divergence approximation based on classification [35, 36], a smooth k-Nearest Neighbor test [37], and the energy statistic [38]. Details of the derivations of these measures can be found in Appendix A.

## 5 Experimental Results

We evaluate our method in a simulated 2D planar navigation problem and manipulation experiments both in simulation and on a physical robot. For all experiments, the goal set planner is run in MPC-style, where optimization is performed at each timestep, then the first action of the lowest-cost trajectory is executed.

**Baselines.** Each baseline involves inference using SVGD [10, 11] with the same running costs, but with different heuristic terminal costs based on previously published work, as defined below.

– *Closest Point:* The nearest goal sample to the start configuration is selected as the goal. The terminal cost is the squared Euclidean distance to this point. A *dynamic* version of this cost is obtained by recomputing the closest point at the beginning of each planning timestep [18].

– *True Goal Distribution:* Given the true goal distribution, the objective is formulated as maximizing the density of the goal distribution at the terminal point [5]. This method is only applicable to the 2D planar navigation task when the goal distribution is known.

– *Mixture Model:* We define a goal likelihood with a Gaussian mixture model obtained by placing each component centered on each goal sample, with a fixed covariance, akin to a kernel density estimator. The goal loss is then the cross entropy over the mixture model, which is equivalent to maximizing the likelihood [5].

**Planar Navigation.** We implement a 2D planar navigation task to characterize the behavior of our approach. We define goal distributions over 2D position for each scene using a parametric distribution (Gaussian, mixture of Gaussians, or uniform). The goal set supplied to the planner consists of a random set of goal samples drawn from the distribution.

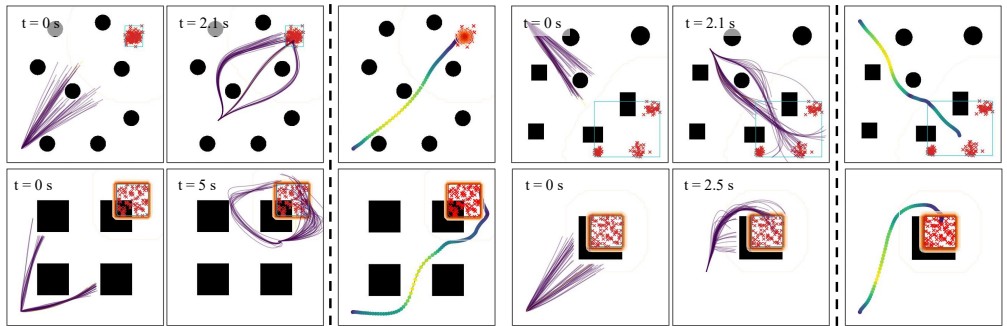

**Figure 5:** Trajectory distributions for the Goal Set Planner using the MMD terminal loss are represented as a set of samples (drawn in purple). The final trajectory is shown in the last pane for each environment, colored with respect to the velocity (yellow is fast, purple is slow). The red points represent the goal samples and the orange contours represent the true goal distribution.

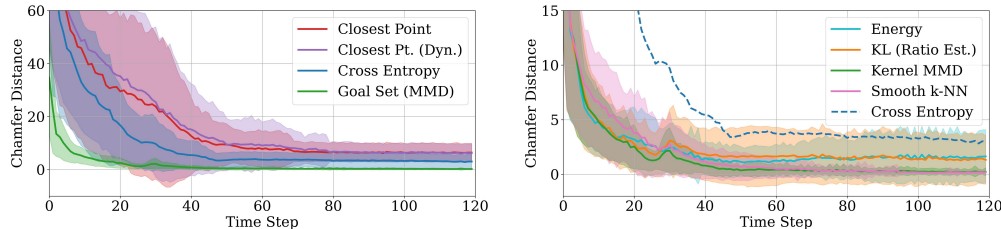

**Figure 6:** Chamfer Distance at each iteration for different terminal costs. (left) Comparison between the baseline methods and our goal set planner with MMD terminal loss. (right) Comparison between different terminal costs. The best performing baseline (Cross Entropy) is shown for reference.

To evaluate the goal set planner, we execute 10 runs per method per scene across 5 scenes. We measure the Euclidean distance between the terminal state in the trajectory and the nearest goal sample for each run. We present the pass rate for each method in Figure 3, where pass rate is defined as the percentage of runs for which the error is less than a given error threshold. All methods achieve similar path lengths (see results in Appendix B.1). The cross entropy baseline performs close to the set-based methods with respect to pass rate, but is prone to falling into local minima. The closest point baselines have a high failure rate on challenging environments, particularly those with collisions near the goal (see Figure 4), or those with the closest point in collision (see bottom-left in Figure 5). While it would be trivial to check for collisions with the closest point in the planar environment, we avoid this additional domain-dependent pre-processing step to better replicate environments in which collisions are expensive or intractable to compute.

Figure 5 shows example trajectory distributions for the goal set planner using the MMD cost. Both the Smooth k-NN and the Kernel MMD terminal losses display moment-matching behaviour, which we posit allows them to be more robust to challenging environments. We posit that the kernel embedding in the MMD loss enables more flexible representation of the goal region compared to the other losses. We select MMD for all manipulation experiments.

*Distribution matching:* We hypothesize that our goal set planning achieves higher success rate in challenging environments due to its ability to better approximate the underlying goal distribution. To verify this, we measure the Chamfer distance between the terminal states in the trajectory distribution at each timestep and the goal samples. The results are shown in Fig. 6. The cross entropy baseline, using the true goal distribution, achieves lower Chamfer distance than the other baselines, but performs worse than the goal set planner due to its mode-seeking behavior. Kernel MMD and smooth k-NN terminal set losses perform better on this metric than the energy statistic and the KL approximation.

**Robotic Grasping.** Stable grasping of arbitrary objects is a challenging problem since the goal distribution consisting of stable grasp poses is intractable to model. We evaluate our goal set planner

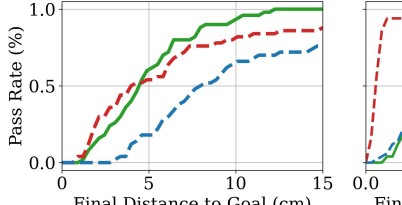
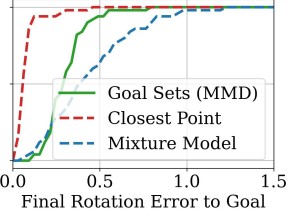

| | Success (%) |
|---|---|
| Closest Pt. | 50 |
| Mixture Model | 40 |
| Goal Set (MMD) | 54 |

**Figure 7:** Pass rate data for the grasping experiments: the Euclidean distance (left) and the L2-norm of the rotation matrix (right) between the final end-effector orientation and the goal.

**Table 1:** Grasp success rate

on a grasping task using grasp samples generated in simulation [6]. We compare our method to the Closest Point and Mixture Model baselines.

We assume that the environment and target object are fully observed. Experiments are performed using the Franka Panda manipulator in the Isaac Gym simulator [39]. We use the STORM library for GPU-accelerated, fully differentiable dynamics and costs [40]. The goal set is represented by $N = 100$ samples of valid grasp configurations drawn from the Acronym dataset [6]. The goal set planner is used to plan to a pre-grasp pose to offset grasp locations in an MPC framework over a fixed horizon. The goal set planner optimizes over all goal samples at each planning iteration, and does not require pre-processing the set prior to planning. After convergence, we use inverse kinematics (IK) to move to the nearest grasp sample, and then lift the object.

We evaluate over 5 runs across 10 scenes containing 8 different objects and 5 environments. Grasp success results are shown in Table 1. We observe that a primary source of failure is due to failure in the IK stage, when moving from the pre-grasp pose to the grasp pose. IK errors can arise from collisions or singularities encountered during the grasp. To better understand the performance of the goal set planner independent from the grasp stage, we measure the planning success rate of our planner to the pre-grasp pose, computed as the distance from the terminal end-effector pose to the nearest goal sample. We present the pass rate for each method in Figure 7. The goal set planner achieves a higher pass rate at thresholds over approximately 4 cm. The closest point method may select goal points which are in collision in some of the example scenes, increasing the distance error, which also accounts for the discrepancy in grasp success. The rotation error is lower for closest point because it matches the orientation of the goal consistently, including for unreachable poses. We posit that the mixture model poorly approximates the true goal distribution explaining its failure.

**Robot Demonstration.**   We demonstrate that our simulated manipulation experiments are transferable to the real robot, under the assumptions of known object and obstacle locations and geometries, through a robotic grasping demonstration. Figure 8 shows two successful grasping runs of the planner, one in an environment without obstacles and one with obstacles. Goal samples are drawn from positive stable grasp examples from the Acronym dataset [6].

**Placement.**   We posit that our planner is more effective than point selection heuristics at dealing with multi-modal goal regions and changing environments due to its ability to consider multiple possible goal regions within the trajectory distribution. To examine this claim, we design a table setting experiment where mugs are placed sequentially on a surface using demonstrations of valid configurations, inspired by demonstration-based arrangement tasks [4]. The goal region consists of

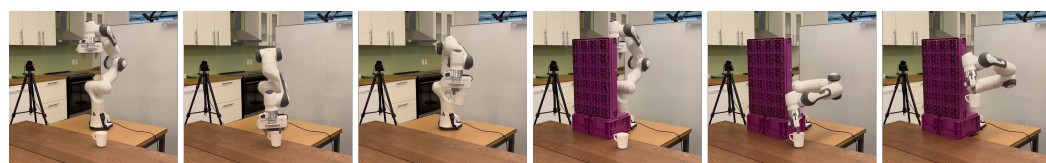

**Figure 8:** Execution runs of the goal set planner on the real robot platform in an environment with no collisions (left three) and with collisions (right three). The robot grasps a mug by planning over a set of successful sample grasp poses obtained from simulation [6]. Trajectories are planned offline and executed on the robot.

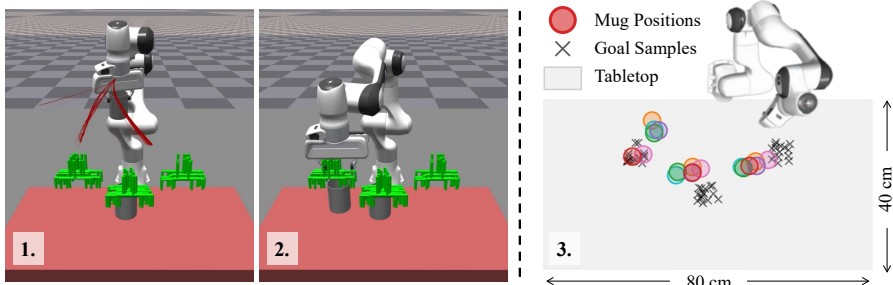

**Figure 9:** Given example demonstrations of mug placement as the goal set (green), the robot plans a path to place a mug. The trajectory distribution (red) considers both free goal distribution modes, and avoids the mode which has a collision due to the first mug (1), ultimately selecting a free mode (2). The final positions of the mugs is show relative to the goal samples over 6 runs (3). The mug colors correspond to runs.

three uniform distributions representing valid terminal gripper 3D positions (see Figure 9, left). We sample 60 points to form the goal set at the beginning of the task, and keep the goal set constant until all mugs are placed. Each subsequent placement operation requires the consideration of a new obstacle in the scene. We employ the MMD terminal loss over the goal samples compared to the 3D terminal positions of the end effector.

We perform six runs for this task, each involving placing three mugs. Figure 9 shows an example trajectory distribution for the goal set planner when placing the second of 3 mugs, which shows the multi-modality of the trajectory distribution. The final mug positions (Figure 9, right) are clustered corresponding to the modes of the demonstrations. We observe that the final mug placements may lie outside the sampled region. This can be caused by early termination, or in some cases, trajectory distributions which converge outside the sampled region. The latter could be caused by singularities in the robot kinematics or local minima.

## 6    Discussion & Conclusion

In this work, we present a formulation for a goal set planner, which considers the planning problem in which an uncertain goal region is represented as a set of samples. The planner is generalizable across goals which can be represented by implicit distributions, without the need for domain-specific, user-defined heuristics. We find that independent of the specific loss used, the GBI formulation outperforms previously proposed approaches for planning to goal sets and distributions. Our method outperforms all baselines on a 2D navigation problem and yields improved plan-to-grasp success on a robotic manipulation platform.

In order to remain true to the general planning problem, our method does not consider domain-specific techniques commonly used for the experiments considered. Our grasping results would likely be improved by integration of pre-planning steps for checking reachability or collisions [41], or pre-grasping refinement of the final pose [31]. We hypothesize that combining these approaches with goal set planning for the application of grasping would improve robustness.

**Limitations:** Maintaining modes in the trajectory distribution can lend added robustness in challenging environments, but it also raises challenges in selecting a single trajectory to execute. We attribute the larger errors in the final pre-grasp pose of our method to the challenge of detecting convergence given the multi-modality of the goal set planner. The terminal set losses considered work best as local measures, meaning gradient information is noisy if trajectories terminate too far from the goal. We mitigate this by applying smooth box priors around the goal samples. Furthermore, the terminal point is not guaranteed to be close to one of the points in the set. This could be mitigated by switching to a point-based cost once close to the goal set. To deploy our planner in real-world applications, more investigation is needed to explicitly handle perceptual uncertainty.

**Acknowledgments**

This work was completed in part while the first author was an intern at the NVIDIA Robotics Lab. We thank the reviewers for their insightful comments. We thank Dieter Fox and Chad Jenkins for helpful discussions and feedback.

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

# Appendices

## A Terminal Losses

This section provides detailed derivations of the different terminal loss functions, $\mathcal{L}_{\text{goal}}(\tilde{X}_T, \tilde{G})$, compared in our generalized Bayesian inference planner. Recall, the loss function $\mathcal{L}_{\text{goal}}(\tilde{X}_T, \tilde{G})$ measures the discrepancy between the terminal state distribution and the distribution of goal states, represented implicitly by samples $\tilde{X}_T$ and $\tilde{G}$ respectively.

**Kernel Maximum Mean Discrepancy (MMD).** The squared MMD [34] measures discrepancy between two distributions $p(x)$ and $q(y)$, and is defined as:

$$\text{MMD}^2(p, q) = \mathbb{E}[k(x, x')] - 2\mathbb{E}[k(x, y)] + \mathbb{E}[k(y, y')] \tag{12}$$

where $k(\cdot, \cdot)$ is a kernel function. In the case where distributions $p$ and $q$ are implicit, an unbiased two-sample approximation of the squared MMD is given by:

$$\text{MMD}_u^2(p, q) = \frac{1}{m-1} \sum_{i=1}^{m} \sum_{j \neq i}^{m} k(x_i, x_j) - \frac{2}{mn} \sum_{i=1}^{m} \sum_{j=1}^{n} k(x_i, y_j) + \frac{1}{n(n-1)} \sum_{i=1}^{n} \sum_{j \neq i}^{n} k(y_i, y_j) \tag{13}$$

where $x_i \sim p$, $1 \leq i \leq m$ and $y_i \sim q$, $1 \leq i \leq n$. When applied to our setting we simply define $\mathcal{L}_{\text{goal}}(\tilde{X}_T, \tilde{G}) = \text{MMD}^2(\tilde{X}_T, \tilde{G})$ where $\tilde{X}_T = \{x_T^{(i)}\}_{i=1}^{M}$ correspond to samples from the terminal state associated with the corresponding trajectory particle $\tau^{(i)}$.

**KL Divergence.** A differentiable two-sample KL divergence approximation can be obtained through density ratio estimation via classification [35, 36]. The KL divergence between distributions $p$ and $q$ is defined as:

$$D^{\text{KL}}(p, q) = \mathbb{E}_{x \sim p(x)} \left[ \log \frac{p(x)}{q(x)} \right]. \tag{14}$$

This quantity can be approximated over samples $x^{(i)} \sim p(x)$ if the ratio $r(x) = p(x)/q(x)$ can be computed analytically:

$$\mathbb{E}_{x \sim p(x)} [\log r(x)] \approx \frac{1}{M} \sum_{i=1}^{M} \log r(x^{(i)}) \tag{15}$$

In the case where the goal distribution and posterior distribution are implicit, the ratio $r(x)$ cannot be evaluated. Instead, the ratio can be approximated through classification, where binary label $y = 1$ indicates that a sample $x$ was drawn from $p$ and $y = 0$ indicates a sample was drawn from $q$. The ratio can then be approximated as:

$$r^*(x) = \frac{P(x|y=1)}{P(x|y=0)}, \tag{16}$$

which can be further simplified to:

$$r^*(x) = \exp(\sigma^{-1}(P(y=1 \mid x))). \tag{17}$$

This approximation reduces the density ratio estimation to a classification problem. Thus, the ratio is computed by evaluating a binary classifier for each sample which can be performed efficiently.

Intuitively, this approach assumes that the KL divergence should be highest when the classifier cannot distinguish between the samples from each distribution. The classifier must be trained individually for each estimated trajectory distribution in order to use this divergence. To improve computational efficiency in practice, the classifier can be initialized with the result from the previous iteration. Finally, we can define $\mathcal{L}_{\text{goal}}(\tilde{X}_T, \tilde{G}) = D^{\text{KL}}(\tilde{X}_T, \tilde{G})$ where the KL divergence is approximated following Eq. (15).

**Smooth K-Nearest Neighbor.** It is possible to define a differentiable two-sample test based on the well known k-NN algorithm as demonstrated in [37]. The Smooth K-Nearest Neighbor test

possesses important statistical properties such as consistency and convergence of its statistics to $f$-divergence. This is despite the complexity of having to solve a combinatorial optimization problem (nearest neighbor match) required by the k-NN method. Let $n_1$ be the number of samples of $p$, and $n_2$ be the number of samples of $q$. The k-NN divergence can be defined as

$$D_\alpha^{\text{NN}} = \int \frac{\alpha^2 p^2(x) + (1-\alpha)^2 q^2(x)}{\alpha p(x) + (1-\alpha)q(x)} dx, \tag{18}$$

for $n_1/(n_1 + n_2) \to \alpha \in (0, 1)$.

As proved in [42], the statistic

$$1 - \frac{T(X_1, X_2)}{(n_1 + n_2)^k}, \tag{19}$$

where $T(X_1, X_2)$ refers to the number of edges connecting samples in a set $X_1 = \{x_i\}_{i=1}^{n_1}$ to a set $X_2 = \{x_i\}_{i=1}^{n_2}$ from a $k$-neighborhood graph created with points in $X_1$ and $X_2$, converges in probability to the $D_\alpha^{\text{NN}}$ divergence, and can be used as an efficient approximation. To make the computation differentiable, the authors of [43] define

$$T(X_1, X_2) = \sum_{i=1}^{n} \sum_{j=1}^{n} s_i(\{x_m\}_{m=1}^{n_1+n_2})_j, \tag{20}$$

where $s_i(\{x_m\}_{m=1}^{n_1+n_2})$ denotes the `softmax` function computed on the Euclidean distances between all points, except point $i$. As the `softmax` function is differentiable, the statistic in Eq. (19) becomes differentiable and can be used directly as our loss function $\mathcal{L}_{\text{goal}}(\tilde{X}_T, \tilde{G})$. To avoid specifying a particular value for $\alpha$, our implementation computes the statistic for several values and averages them as the final result.

**Energy Statistic.** This two-sample test is based on Newton's gravitational potential energy which relates two entities by the Euclidean distance between them [38]. Given two distributions $p(x)$ and $q(y)$, the energy distance is defined as:

$$D^{\text{E}}(p, q) = 2\mathbb{E}[||x - y||^2] - \mathbb{E}[||x - x'||^2] - \mathbb{E}[||y - y'||^2], \tag{21}$$

where $x$ and $y$ are independent random variables. The corresponding two-sample statistic given two sets of samples $X = \{x_i\}_{i=1}^{n_1}$ and $Y = \{y_j\}_{j=1}^{n_2}$ can then be written as:

$$D^{\text{E}}(X, Y) = \frac{2}{n_1 n_2} \sum_{i=1}^{n_1} \sum_{m=1}^{n_2} ||x_i - y_m||^2 - \frac{1}{n_1^2} \sum_{i=1}^{n_1} \sum_{j=1}^{n_1} ||x_i - x_j||^2 - \frac{1}{n_2^2} \sum_{l=1}^{n_2} \sum_{m=1}^{n_2} ||y_l - y_m||^2. \tag{22}$$

This provides a computationally efficient statistic which can be directly used as our loss $\mathcal{L}_{\text{goal}}(\tilde{X}_T, \tilde{G}) = D^{\text{E}}(\tilde{X}_T, \tilde{G})$.

## A.1 Practical Considerations

The statistics considered in this work are good local approximations of distribution divergences. In the case of trajectory optimization, when the terminal states in early planning iterations are far from the goal set, the goal loss gradients can be uninformative. We therefore include a prior in our set planning method consisting of a smooth uniform distribution constructed by placing a bounding box around the goal samples. This can be included in our framework by multiplying a prior over the terminal state $p(x_T)$ with the goal likelihood in Eq. (6). This mitigates the poor divergence approximation in early iterations. Note that the uniform prior is insufficiently informative on its own, particularly in cases where the goal set is multi-modal. Furthermore, the prior needs to be differentiable, which is not the case for a standard uniform distribution. Therefore we define a smooth uniform prior in the region $R = x_T : a \leq x_T \leq b$ as

$$p(x_T) \propto \exp\left(-d(x_T, R)^2 / \sqrt{(2\sigma^2)}\right) \tag{23}$$

where $d(x, R) = \min |x - x'|$, $x' \in R$ is a distance function, and $\sigma$ controls the *sharpness* of the approximation.

Once inference over the trajectory distributions converges, we must select a single trajectory estimate to execute. A common approach to accomplish this is by taking the mean, or weighted mean, of the particle set. This method is ineffective when the trajectory distribution is multi-modal. An alternative approach is to pick the maximum weighted particle. Our proposed set-based terminal losses yield a single score over the whole distribution, which does not enable weighing individual particles based on terminal loss. To select our final sample, we instead select the lowest cost trajectory. In practice, we also include the prior in the weight computation to avoid local minima with very low-cost trajectories.

# B  Experiment Details

## B.1  Planar Navigation

The agent state $x_t$ is composed of a 2D position and velocity, and the control signal $u_t$ is a 2D acceleration. We use known, linear dynamics in a fully observed environment for the controller rollouts.

**Losses.**  For each method, the running cost is summed over each timestep, where the cost for one timestep is:

$$c_t(x_t, u_t, z) = x_t^\top Q x_t + u_t^\top R u_t + \alpha \, c_{\text{SDF}}(x_t, z) \tag{24}$$

where $Q$ and $R$ are quadratic cost parameters for the state and action, and $c_{\text{SDF}}(x_t, z)$ is the obstacle avoidance term, computed using the Signed Distance Function (SDF) over the environment $z$. The goal loss $\mathcal{L}_{\text{goal}}(\tilde{X}_T, \tilde{G})$ is a set goal loss which is differentiable with respect to the trajectory $\tau$.

**Implementation Details.**  For each of our goal set planner ablations, $N = 50$ samples are randomly selected from goal distribution, except for *KL (Ratio Estimation)*, which uses $N = 100$ samples. This method involves training a learned classifier so is aided by a higher sample size. For the closest point methods, the goal sample with the smallest Euclidean distance from the start state is selected. For all the methods, $M = 50$ particles are used to represent the trajectory distribution. The particles represent the discrete control signals, $u_t$, which are 2D accelerations at each 0.1 second timestep over a horizon of 3 seconds. All planners are initialized with the distribution from the previous timestep, shifted to the current timestep, and run for 50 iterations. We use the Adam optimizer [44] to select the step size in the SVGD update rule.

The KL divergence uses a 3 layer fully-connected network as the classifier, retrainined at each timestep. To mitigate computational complexity, we warm start the training with the weights from the previous timestep. The Kernel MMD uses an RBF kernel, with a bandwidth selected by applying the median heuristic over the goal samples [44]. The Smooth k-NN loss uses a value of $k = 1$.

We use the RBF kernel for SVGD, and set the bandwidth using the median heuristic, a popular technique for choosing the kernel bandwidth which yields a good estimate in many cases [45]. Without access to data consisting of trajectory samples, we assume that they are normally distributed with covariance $\sigma = 1$. Under these assumptions, it follows that the expected distance between samples drawn from the distribution is $2D$, where $D$ is the trajectory dimension.

**Extended Results.**  We visualize the total Euclidean path length for each method in Figure 10. The baselines are shown with hatched bars. We exclude paths for which the terminal state is not within 40 cm of any goal sample. All methods achieve similar path lengths, with the closest point baselines consistently resulting in slightly shorter paths. This is unsurprising given the distance-based goal selection bias, but comes at the cost of being less robust to different environments.

## B.2  Grasping

Figure 11 shows a case where the goal set planner finds a reachable point from the set, but the closest point method may select goal points which are in collision in some of the example scenes. This phenomenon helps explain the discrepancy in grasp success at the cost of increasing distance error.

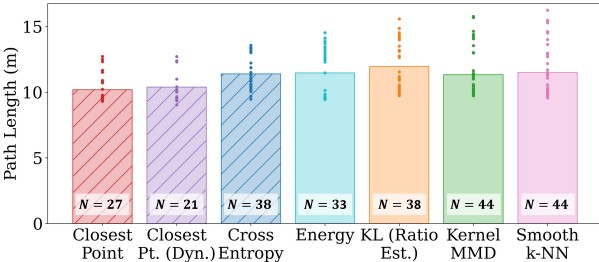

**Figure 10:** Average path length for each terminal cost for the planar navigation task. The points indicate results for individual runs. Only successful trajectories are included, where success is defined as getting to within 40 cm from any sample within the goal set. The numbers on the bars indicate the number of successful runs out of 50 in each category.

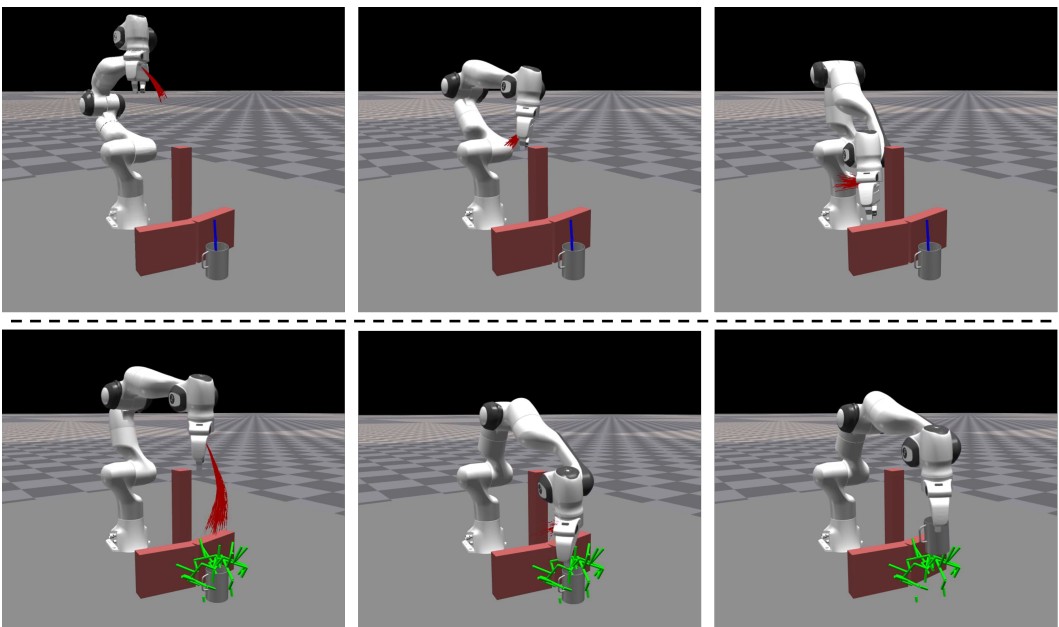

**Figure 11:** Example failure execution for the closest point planner (top) compared the the goal set planner (bottom). The closest point selected is shown in blue. The point is not reachable in the environment, causing the robot to fail to reach it (top right). The set planner considers all the grasp samples, shown in green. It finds a reachable grasp point and grasps successfully (bottom right).

