# OpenReview forum: "Ready, Set, Plan! Planning to Goal Sets Using Generalized Bayesian Inference"
_robot-learning.org/CoRL/2023/Conference — CoRL 2023 Poster_

### Official Review · Reviewer_aNNG · 2023-07-20

**Confidence:** 4
**Originality:** Good
**Technical Quality:** Excellent
**Clarity Of Presentation:** Excellent
**Impact:** 3

**Recommendation:**

Weak Accept: I recommend accepting the paper, but will not argue for my recommendation if the majority of other reviewers have a different opinion.

**Review:**


This is a very good problem formulation.  Learning a set of good solutions or a scoring functions makes the results of this module composable with other modules downstream, for example.  It uses a very nice overall objective that optimizes a property of the ensemble

The basic experiments are well designed and illuminating.   But I'm confused about the robot grasping experiments.  I would have
thought you would be planning the whole trajectory.  But if that's true, you wouldn't need IK, would you?

Small points:
- Sefine the C is the terminal cost function
- What is z (the environmental observation)?  how is that different from x?   I guess it's the static part of the state?
- Do you assume deterministic dynamics?  I assume so if you're talking about a minimum cost trajectory.  But then we don't need p(x_t+1 | x_t, y_t)  --- it can just be a function
-  I have never understood the attraction of writing a collision constraint in the form of equation (5), but okay
- In figure 4, in the second frame of each batch, why does the starting point seem different from the first frame?
- Why don't we have 100% pass rate ever?  Are some instances  infeasible?  Are we not running the optimization long enough?

I suspect that this might all be clearer just written as one straight-up optimization problem without going via the probabilistic
formulation (since really there is nothing here that is inherently probabilistic). I guess this formulation makes a useful set of
pre-existing tools (like SVGD) available, so that's good.

**Quality Of The Limitations Section:**

Limitations are addressed clearly

**Questions For Rebuttal:**

See above

**Robotics Focus:**

Sufficient demonstration on hardware

**Summary Of Paper:**

A method for plan to achieve goals that are specified only in terms of a set of samples of desired trajectory end-points.   The objective is articulated as a probability distribution and Stein variational gradient descent is used to optimize a set of particles to have low divergence with respect to the target distribution.  The method is illustrated in a 2D pedagogical domain and for a grasping robot, simulated and real.

**Summary Of Recommendation:**

The method is very nice and well described.  However:
- It would be good to have a real use-case for this method described:   for example, it seems more likely that I would have available an expensive (possibly noisy) grasp evaluation function (either something analytic based on a mesh or a neural net) than a set of demonstrations.
- The experiments are kind of worrisome because it wasn't clear why we only had about a 50% success rate.  I can think of lots of reasons why that might be, but some actual error analysis seems important in order to really evaluate the underlying strengths and weaknesses of the method.

---

### Official Review · Reviewer_Jat5 · 2023-07-20

**Confidence:** 4
**Originality:** Very Good
**Technical Quality:** Very Good
**Clarity Of Presentation:** Good
**Impact:** 4

**Recommendation:**

Strong Accept: I recommend accepting the paper and will argue for my recommendation even if other reviewers hold a different opinion.

**Review:**

First, the paper is well introduced and nicely written. The general ideas and the technical details of the method are presented well. The approach is sound and the property that only samples from a goal set are required are very appealing. The experiments are generally carried out well (although see my comments below for possible improvements) and demonstrate the effectiveness of the approach. The video in the supplementary material gives a good overview of the method and its core ideas.

The only two points that could be improved are the following:

- The structure of Section 4 could be improved. In particular, the paragraphs Generalized Bayesion Inference and Stein Variational Inference feel like they should be moved to the Related Work Section. The current structure disconnects the beginning of Section 4 and the paragraph "Terminal Losses for Goal Sets"

- While the experiments are carried out well, I would have liked to see a more diverse set of experiments. For instance, what happens when the problem consists of disconnected goal sets (e.g. a robot having to place a cup on one of two tables)? Does the method still work? My intuition is that it should still work, but this should be demonstrated

**Quality Of The Limitations Section:**

Limitations are addressed clearly

**Questions For Rebuttal:**

- It is not 100% clear to me how the gradient in eq.(11) is derived. In eq.(4) you have the goal likelihood $p_{goal}(\tilde{G} | x_T)$ that seems to be expressed in terms of the variational term $q_{\mathcal{L}}(x_T | \tilde{G})$ in eq.(10). It would be great if some additional details were provided (perhaps a via step-by-step derivation in the Appendix) how exactly eq.(11) is derived.

**Robotics Focus:**

Sufficient demonstration on hardware

**Summary Of Paper:**

This paper proposes a goal set planner for planning problems in which goal regions are difficult to express explicity. At its core, the method frames the planning problem as a Bayesian inference problem and combines methods from generalized Bayesian inference and Stein variational inference to construct a differentiable loss function. Optimizing this loss function with respect to a set of sampled trajectories results in trajectories that reach an arbitrary goal set. Crucially, instead of expressing the goal set explicitly, the proposed method only requires samples from the goal set. The method is evaluated on set of experiments involving simulated 2D navigation and manipulation task, and a grasping task using a physical robot.

**Summary Of Recommendation:**

In summary, the paper proposes a trajectory-optimization based planner that is designed to handle arbitrary goal sets that are only represented via samples. The method includes novel and interesting ideas that I think are of interest to the wider robotics community.

---

### Official Review · Reviewer_CHoH · 2023-07-30

**Confidence:** 4
**Originality:** Fair
**Technical Quality:** Good
**Clarity Of Presentation:** Very Good
**Impact:** 3

**Recommendation:**

Weak Accept: I recommend accepting the paper, but will not argue for my recommendation if the majority of other reviewers have a different opinion.

**Review:**

The paper is well written and the proposed method was evaluated in both simulation and real world setup.

Here are a few of my concerns:

1) The evaluation metrics covered a couple of things, including "final distance to goal" shown in Figure 7, as well as grasp success rate shown in Table 1. While the former metrics could cover both navigation of mobile robot and robot arm manipulation, the latter on is more focused on manipulation related applications. Is the navigation to the goal set (or grasp pose) the major contributing factor of success rate on manipulation? It could be better if the author could clarify on it a bit.

2) The proposed approach assumed "goal set" however in Figure 4 (top right) the planning seems stopped at the bounding box of the all goals. Is this cutting off a bit too early or expected behavior? I was wondering if the planner would return multiple trajectories with some topological variations or it will pick the best of out of several candidates.

3) In the real world experiments (Figure 8) seems only one cup is the target, not sure how the "goal set" in this case looks like. Could be really interesting to visualize it (like Figure 9 does, though Figure 9 does not have the obstacle setup in Figure 8), and I was wondering if some of the target pose in the "goal set" would require the robot arm to reach the cup from the other side of the obstacle.

**Quality Of The Limitations Section:**

Additional details required

**Questions For Rebuttal:**

Please refer to my questions in the "review" section.

**Robotics Focus:**

Highly relevant to robotics but no hardware experiments

**Summary Of Paper:**

The paper proposed an planning algorithm to a goal set with bayesian inferences. The proposed approach was evaluated in simulation first and then on a robot arm in a offline manner.

**Summary Of Recommendation:**

I recommend weak reject, and since I don't have a lot of the experiences on robot manipulation I could be naive. The paper seems more likely to be looking at manipulation applications so I am happy to update my review if there are more experienced review working on manipulation.

---

### Official Review · Reviewer_bCZS · 2023-07-31

**Confidence:** 3
**Originality:** Fair
**Technical Quality:** Fair
**Clarity Of Presentation:** Fair
**Impact:** 2

**Recommendation:**

Weak Accept: I recommend accepting the paper, but will not argue for my recommendation if the majority of other reviewers have a different opinion.

**Review:**

Strengths
- The problem setup is quite interesting and practical
- Clear illustration of the problem setup
- Real robot experiments

Weaknesses
- Experiments are extensive but the trends are a bit underwhelming. Not much of a significance difference compared to a bit simple baseline in 2D environments, and almost no significant difference in success rates for grasping experiments.
- Robot experiments are nice but it's difficult to understand the main message of those experiments. What is the main challenge for these experiments and what is the main point of demonstrating that robots can work with the proposed planner? How do other baselines work for this and how would they fail?
- This is a bit minor but also important point. I tried my best to understand the paper, but it feels like the paper is difficult to parse as it's not self-contained. The paper assumes a lot of background on multiple components and provide background in a little depth, e.g., planning as inference, stein variational inference, kernel MMD, Champer distance, etc. Formulations in Section 3 are also a bit dense to parse. Maybe adding a more intuitive explanation and also the detailed formulation (at least in the supplementary material) could help readers understand the main idea.

**Quality Of The Limitations Section:**

Additional details required

**Questions For Rebuttal:**

- Please address my points in Weaknesses
- What is the exact definition of pass rate in experiments? Also, should it be 0\~100% instead of 0.0\~1.0?
- Please add more labels and legends to figures, especially Figure 4 is very difficult to parse without reading the main text very carefully.

**Robotics Focus:**

Relevant but unlikely to deploy to hardware in near future

**Summary Of Paper:**

This paper presents a new method that plans towards a set of (sampled) goals instead of a specific goal chosen with heuristics. The main idea is to formulate the problem as a generalized bayesian inference, which allows for making an inference without assuming the explicit distribution for goals. Specifically, the divergence between the goal distribution and terminal state distribution can be evaluated using the samples from each distributions. Then SVGD is used for inference that enables us to avoid evaluating the loss for each particle corresponding to each terminal state. Experiments are conducted on simple 2D planar environments, also on simulated grasping environments and real-world environments.

**Summary Of Recommendation:**

I'd like to recommend the paper to be 'weakly rejected'. In its current status, the experimental results are not sufficient to support the benefit over baseline heuristics and the main message of several experiments are not crystal clear. Moreover, the paper have a room for improvement in terms of clarity, by making it be more self-contained and be more intuitive about explaining the main method.

---

I have updated my score to 'weak accept' per the discussion during the rebuttal period.

---

### Decision · Program_Chairs · 2023-08-30

**Decision:**

Accept (Poster)

**Comment:**

The paper presents a strategy for goal-set planning, for problems in which the goal region is difficult to specify using Bayesian inference. Before the rebuttal, the reviewers were somewhat confused about the robotics experiment and its significance, although they found the motivation and method convincing. The ambiguities were resolved and all the reviewers have agreed to accept the paper.